# A Novel 10-Base Pair Deletion in the First Exon of *GmHY2a* Promotes Hypocotyl Elongation, Induces Early Maturation, and Impairs Photosynthetic Performance in Soybean

**DOI:** 10.3390/ijms25126483

**Published:** 2024-06-12

**Authors:** Xiaobin Zhu, Haiyan Wang, Yuzhuo Li, Demin Rao, Feifei Wang, Yi Gao, Weiyu Zhong, Yujing Zhao, Shihao Wu, Xin Chen, Hongmei Qiu, Wei Zhang, Zhengjun Xia

**Affiliations:** 1Key Laboratory of Soybean Molecular Design Breeding, State Key Laboratory of Black Soils Conservation and Utilization, Northeast Institute of Geography and Agroecology, Chinese Academy of Sciences, Harbin 150081, China; ca_zhuxiaobin@163.com (X.Z.); haiyanwang2001@163.com (H.W.); lilimonum@163.com (Y.L.); wangfeifei@iga.ac.cn (F.W.); gaoyi181@mails.ucas.ac.cn (Y.G.); zhongweiyu0312@163.com (W.Z.); yujingzhao99@163.com (Y.Z.); 2University of Chinese Academy of Sciences, Beijing 100049, China; 3Soybean Research Institute, Jilin Academy of Agricultural Sciences, Changchun 132102, China; rdm1397155464@163.com (D.R.); qhm2001-2005@163.com (H.Q.); zw.0431@163.com (W.Z.); 4Institute of Industrial Crops, Jiangsu Academy of Agricultural Sciences, Nanjing 210014, China; adidaswsh@yahoo.com (S.W.); cx@jaas.ac.cn (X.C.)

**Keywords:** soybean, *Gmeny*, phytochromobilin, hypocotyl elongation, photosynthesis, GA signaling, maturity

## Abstract

Plants photoreceptors perceive changes in light quality and intensity and thereby regulate plant vegetative growth and reproductive development. By screening a γ irradiation-induced mutant library of the soybean (*Glycine max*) cultivar “Dongsheng 7”, we identified *Gmeny*, a mutant with elongated nodes, yellowed leaves, decreased chlorophyll contents, altered photosynthetic performance, and early maturation. An analysis of bulked DNA and RNA data sampled from a population segregating for *Gmeny*, using the BVF-IGV pipeline established in our laboratory, identified a 10 bp deletion in the first exon of the candidate gene *Glyma.02G304700*. The causative mutation was verified by a variation analysis of over 500 genes in the candidate gene region and an association analysis, performed using two populations segregating for *Gmeny*. *Glyma.02G304700* (*GmHY2a*) is a homolog of *AtHY2a* in *Arabidopsis thaliana*, which encodes a PΦB synthase involved in the biosynthesis of phytochrome. A transcriptome analysis of *Gmeny* using the Kyoto Encyclopedia of Genes and Genomes (KEGG) revealed changes in multiple functional pathways, including photosynthesis, gibberellic acid (GA) signaling, and flowering time, which may explain the observed mutant phenotypes. Further studies on the function of GmHY2a and its homologs will help us to understand its profound regulatory effects on photosynthesis, photomorphogenesis, and flowering time.

## 1. Introduction

Light is the energy source for photosynthesis. Plants can perceive changes in light quality and intensity through photoreceptors and can respond to these environmental signals by regulating their photomorphogenesis, growth, and reproductive development. In general, under dense planting conditions, plants compete for light and grow tall, but also become spindly and prone to lodging. This response, known as the shade-avoidance syndrome [1], is induced by specific aspects of the light quality under shaded conditions. Red light (R) can be absorbed by the plants, but most far-red light (FR) is transmitted or reflected, resulting in a reduced R/FR ratio under dense planting. There are four main types of photoreceptors in plants: phytochromes (R and FR receptors), UVB receptors, cryptochromes (blue/UVA receptors), and phototropins [2].

Phytochrome apoproteins are encoded by a small multigene family, which comprises five members of *Phytochromes* (*PHYA–PHYE*) in *Arabidopsis thaliana* [3]. In general, phytochromes are proteins of around 125 kDa that exist in two forms: one form absorbs R (Pr, λmax = 660 nm) and the other absorbs FR (Pfr, λmax = 730 nm). Phytochromes are biosynthesized in the inactive Pr conformation and converted to the active Pfr conformation upon absorbing R. Pfr can be converted back to the Pr conformation by the absorption of FR, in a process called photoreversion. Reversion of Pfr to Pr can occur spontaneously in the dark (dark reversion) and is strongly influenced by temperature in both the light and the dark (thermal reversion) [3]. When a plant is exposed to R, the phytochrome changes from the inactive Pr to the biologically active Pfr and enters the nucleus from the cytoplasm. In the nucleus, Pfr interacts with phytochrome-interacting factors (PIFs) and the PHOTOMORPHOGENIC/DE-ETIOLATED/FUSCA (COP/DET/FUS) complex, thereby regulating the expression of genes involved in plant growth and development [2,3,4]. *PHYA* and *PHYB* play the most significant roles [3]. When grown under FR, *phyA* mutants display aberrant skotomorphogenesis (etiolation) [5], whereas under R and white light, *phyB* mutants show an elongated hypocotyl and reduced chlorophyll contents instead of the typical de-etiolation response [6].

Phytochromes detect light via a covalently bound linear tetrapyrrole chromophore (phytochromobilin; 3E-PΦB) and function as a dimer [7]. Each phytochrome apoprotein contains two modules, the N-terminal photosensory module (PSM) and the C-terminal module (CTM), connected by a hinge region [8]. The PSM is mainly responsible for the perception of light signals, while the CTM forms dimers to regulate the nuclear localization of the protein and transmit light signals to downstream components [8]. PΦB is biosynthesized in plastids by a series of enzymatic reactions, a pathway which shares a precursor (5-aminolevulinic acid; 5-ALA) with the biosynthesis of chlorophyll. Heme is converted by heme oxygenase (HO) into biliverdin IX (BV), which is then reduced to 3Z-PΦB by phytochromobilin synthase. 3Z-PΦB and its isomer 3E-PΦB are the functional precursors of the chromophore of phytochromes [9,10]. PΦB is transported into the cytoplasm, where it covalently binds to the cysteine residues of phytochrome apoproteins to form phytochrome holoproteins.

PΦB synthase is the last enzyme in the chromophore biosynthetic pathway and is a member of the ferredoxin-dependent bilin reductase (FDBR) family. Mutations in PΦB synthase affect the biosynthesis and function of the photochrome chromophore [10]. Kohchi et al. (2001) initially reported the long-hypocotyl mutation of the *HY2* gene encoding PΦB synthetase in Arabidopsis [10]. Subsequently, abnormal PΦB synthase function was reported in various other species, including the *pcd2* mutant in pea (*Pisum sativum* L.) [11], the *se13* mutant in rice (*Oryza sativa* L.) [12], the *elh1* mutant in cucumber (*Cucumis sativus* L.) [13], the *Gmlin1* mutant in soybean [14], and the *Mtpφbs* mutant in alfalfa (*Medicago truncatula*) [15]. Studies have shown that chromophore-deficient mutants generally have a light green or yellowish leaf phenotype, accompanied by a reduction in chlorophyll. This can be attributed to the mutation of key enzymes in the PΦB biosynthesis pathway, which cause the accumulation of heme and inhibit the biosynthesis of 5-ALA in a regulatory feedback loop [16].

Phytochromes play important roles in the photoperiodic response of plants, with their disruption altering flowering or maturity times [14,17]. During photomorphogenesis, PIF3 and PIF4 interact with DELLA proteins, negative regulators of gibberellic acid (GA) signaling, highlighting the regulatory interactions between the light and GA pathways [18,19]. Under darkness, an increased GA content can release the inhibitory effect on PIF3 and PIF4 as GA can bind to and degrade DELLA proteins; under light, phyB in the nucleus inhibits GA signaling, allowing DELLA to interact with PIF3 and PIF4 and preventing the expression of downstream skotomorphogenesis genes [20,21].

Soybean (*Glycine max* (L.) Merrill) is an important crop that provides high-quality vegetable protein and oil for human and animal consumption [22]. In this study, we identified a soybean mutant, *Gmeny*, with elongated nodes and yellowing leaves. It was determined to have a decreased chlorophyll content and altered photosynthetic performance. Through the analysis of bulked DNA and RNA data using the BVF-IGV pipeline established in our laboratory, we established that a 10 bp deletion in the first exon of *Glyma.02G304700*, which was the causal mutation for *Gmeny*. *Glyma.02G304700* encodes a PΦB synthase. A transcriptome analysis revealed altered gene expressions related to photosynthesis, GA signaling, and flowering time in the mutant, which may explain the observed physiological and phenotypical changes. Building on previous characterizations of this gene, we revealed that the variations in *Glyma.02G304700* and its homologs can lead to profound changes in leaf color, chlorophyll, and the elongation of the hypocotyl and epicotyl. These insights open a new avenue for dissecting the complex regulatory network of GmHY2a in the control of vegetative and reproductive plant growth.

## 2. Results

### 2.1. The Gmeny Mutant Displays Yellowing Leaves and Elongated Nodes

We generated a mutant library of the soybean cultivar “Dongsheng 7” using gamma ray (60Co) irradiation. Each line of the M_3_ population was derived from a single M_2_ plant. The line T372 showed segregation in leaf color and the hypocotyl and node lengths, with a total of 28 plants showing the WT phenotype and 12 plants displaying the mutant phenotype of yellowing leaves with elongated hypocotyl and nodes. These mutants were subsequently referred to as *Gmeny*. In M_4_, the four lines derived from *Gmeny* plants showed a homozygous *Gmeny* phenotype. Of the 10 lines derived from the WT plants, four showed homozygous WT phenotypes and six displayed segregation, with a total of 111 *Gmeny* plants and 361 WT phenotypes, fitting a 3:1 Mendelian segregation ratio for a single recessive gene. In the F_2_ population of *Gmeny* × Hefeng55, the segregation of 206 *Gmeny* plants and 675 WT plants is also consistent with the 3:1 Mendelian segregation ratio.

When grown in a greenhouse under natural sunlight, the *Gmeny* mutant displayed yellowing leaves and elongated nodes at the vegetative stage (Figure 1A). The hypocotyl length (7.550 ± 0.76 cm) and epicotyl length (8.261 ± 1.032 cm) of *Gmeny* were both significantly greater than those of the WT (5.259 ± 0.793 cm and 5.104 ± 0.739 cm) (Figure 1B,C). At the R2 stage in the natural field, *Gmeny* showed yellowing leaves when compared with the WT (Appendix A). *Gmeny* also matured about 10 days earlier than the WT plants (Appendix A). At the R8 stage, the node number on the main stem and the total number of pods per plant were both significantly lower in *Gmeny* than in the WT; however, *Gmeny* and WT displayed similar plant heights and effective branches (Appendix A). This inconsistency between the different growth characteristics could be attributed to *Gmeny* stopping vegetative growth prematurely but producing fewer but elongated epicotyl and basal nodes and an elongated hypocotyl.

### 2.2. Yellowing Leaves of Gmeny Have Decreased Chlorophyll Contents and Changes in Photosynthetic Capacity

In the field, *Gmeny* leaves had significantly lower levels of chlorophyll a (9.854 ± 0.5190 mg/g of fresh weight), chlorophyll b (2.269 ± 0.182 mg/g), and chlorophyll a + chlorophyll b (12.12 ± 0.692 mg/g) than those of the WT plants (15.530 ± 0.245 mg/g, 4.668 ± 0.101 mg/g, and 20.20 ± 0.195 mg/g, respectively) at *p* < 0.001 (Figure 2A–C). In contrast, the carotenoid content (2.487 ± 0.0956 mg/g) in the leaves of *Gmeny* was lower than in the WT (2.813 ± 0.058 mg/g) (Figure 2D) at *p* < 0.05. Similarly, the total pigment content in *Gmeny* was consistently lower than that of the WT plants at three developmental stages (VC, V1, and V3) (Figure 2E).

The *Gmeny* leaf showed a significantly higher net photosynthetic rate (25.25 ± 0.800 μmol/m^2^/s) (Figure 2F), stomatal conductance (0.4634 ± 0.006 mol/m^2^/s) (Figure 2G), and transpiration rate (10.25 ± 0.098 mmol/m^2^/s) (Figure 2H) than the WT plants (20.54 ± 0.809 μmol/m^2^/s, 0.5644 ± 0.0225 mol/m^2^/s, and 11.430 ± 0.276 mmol/m^2^/s, respectively) at *p* < 0.05. In contrast, the mutant’s internal CO_2_ concentration of 221.4 (±1.739) ppm was not significantly different from that of the WT (217.4 ± 4.305 ppm) (Figure 2I).

In addition to the decreased chlorophyll content, transmission electron microscopy also revealed a noticeable difference in the chloroplast structure between the WT (Figure 2J–L) and the *Gmeny* plants (Figure 2M–O). The thylakoid membranes of the *Gmeny* chloroplasts were not as well organized as in the WT (Figure 2J,K,M,N). The lamellar structure of the *Gmeny* chloroplasts was dark, fuzzy, and had fewer layers when compared with that of the WT (Figure 2L,O). Taken together, these results demonstrate that the chloroplast structure in the *Gmeny* mutant was not fully developed or was disrupted, leading to profound changes in the photosynthetic parameters.

### 2.3. The BVF-IGV Pipeline Identified Glyma.02G304700 as the Candidate Gene for the Gmeny Phenotype

The initial analysis of the resequencing data using MutMap [23] is presented in Appendix A. Due to the limitations of MutMap revealed in this study, BVF-IGV pipeline was subsequently used [24] (Figure 3). Two pools of genomic DNA were extracted from the leaves of 12 plants with the *Gmeny* mutant phenotype and 28 plants with the WT phenotype from the line Dongsheng T372. The 150 bp paired-end resequencing was performed on an Illumina platform at Annoroad Gene Technology (Beijing, China). For the *Gmeny* pool, the total number of reads was 234,497,694, the total number of bases in high-quality (HQ) reads was 34,446,549,900, and the total number of HQ bases in HQ reads was 33,382,537,952. For the WT pools, the total number of reads was 245,738,426, while the total number of HQ bases in HQ reads was 36,096,678,600.

We included only allelic variations annotated as a “missense variation” or “frameshift variation” in the first round of the analysis. A total of 32,869 allelic variations for the *Gmeny* bulk and 30,783 for the WT bulk were identified when compared with the V275 reference genome. By manipulating the VCF data in Excel, we eliminated the common allelic variations between the *Gmeny* and WT bulks, leaving 1427 loci (772 genes) specific to *Gmeny* and 983 loci (383 genes) specific to the WT. These genes were manually assessed individually or in batches by taking snapshots of each gene using the function “Run Batch Script” built into IGV. In our laboratory, this processing pipeline was named the BVF-IGV pipeline [24].

After an initial check, we identified a 10 bp deletion (47,983,485–47,983,494) in *Glyma.02G204700*, with the *Gmeny* bulk showing a homozygous deletion across all reads. The WT bulk displayed a heterozygous sequence, with the 10 bp deletion observed in an average of 9 reads of the 27–28 reads spanning the mutated region. This segregation pattern was in good accordance with the fact that half of the WT plants were heterozygous at *Gmeny* (Figure 3). *Glyma.02G204700* was therefore considered a strong candidate gene for *Gmeny* based on using the BVF-IGV pipeline.

To validate the authenticity of the candidate gene, we examined the polymorphisms in the region of *Glyma.02G204700* between *Gmeny* and the WT. This gene is situated close to the telomere of chromosome 2 and is located on the minus strand; therefore, we investigated 500 genes on the 3′ untranslated region side and all 76 genes on the 5′ UTR side. Although a few variants other than the 10 bp deletion in *Glyma.02G204700* were detected using IGV, the segregation ratio among the reads of the variants between the *Gmeny* and WT bulks greatly deviated from the expected ratio. Furthermore, we analyzed the bulked RNA-seq data taken from the *Gmeny* and homozygous WT plants using the BVF-IGV pipeline. The variants in the candidate gene and other locations further supported the conclusion that *Glyma.02G204700* is the mutated gene in the *Gmeny* mutant (Figure 3). Also, the 10 bp deletion in *Gmeny* was validated by performing PCR and sequencing the resulting products using the primer pairs listed in Appendix A.

We performed an association analysis using phenotype and genotyping data using the 10 bp deletion-derived Eny_HM1 and Eny_L1 primer pairs (Appendix A). To increase the genotyping efficiency, we bulked 10 plants with the *Gmeny* phenotype for genotyping either by direct PCR using the Eny_HM1 primer pair (Appendix A) or by sequencing PCR products amplified from the Eny_L1 primer pair. Of the 40 samples of the T372 line, all *Gmeny* plants possessed the 10 bp deletion genotype, while WT plants were either homozygous WT genotypes or heterozygous. In the M_4_ generation, all tested *Gmeny* plants, either from the homozygous *Gmeny* plants or from six segregation lines, demonstrated a homozygous 10 bp deletion genotype. Of the 361 WT plants tested, 90 showed a homozygous WT genotype, while 271 had heterozygous genotypes.

In the F_2_ population of *Gmeny* × Hefeng55, all 206 *Gmeny* plants possessed the 10 bp deletion, while of the 675 WT plants, 190 displayed homozygous WT genotypes and 485 plants were heterozygous.

### 2.4. Functional Prediction and Evolutionary Analysis of Glyma.02G204700 in Soybean

On the reference genome of Wm82.a2.v1, *Glyma.02G304700* is situated between positions 47,976,140 and 47,983,615 on the reverse strand of chromosome 2. *Glyma.02G304700* was functionally annotated as a *GmHY2a* gene encoding a 1.3.7.4 PΦB synthase with an FDBR domain (Wm82.a2.v1). The full length of *Glyma.02G304700* is 7352 bp, including a 990 bp coding region which encodes 329 amino acids. Notably, *Glyma.02G304800* is located from position 47,980,214 to position 47,980,782 on the forward strand of chromosome 2 and therefore overlaps *Glyma.02G304700*. Physically, *Glyma.02G304800* is positioned within the fourth intron of *Glyma.02G304700* on the opposite strand, but no polymorphism was observed in this region between *Gmeny* and the WT. The 10 bp deletion in the first exon of *Glyma.02G304700* leads to a 12 amino acid change, including a deletion of nine amino acids, in the N-terminal of its protein in the *Gmeny* mutant compared with the WT protein (Figure 4A).

A phylogenetic analysis revealed two highly similar homologs of Glyma.02G304700 in soybean: Glyma.14G009100 (score of 307, similarity of 95.4%) and Glyma.14G136300 (score of 136, similarity of 85.4%). Several homologs were identified in other leguminous species, including Phvul.008G284200 (score of 551, similarity of 90.2%) from the common bean. In the model plants, highly homologous sequences were identified in Arabidopsis (AT3G09150; score of 354, similarity of 76.9%) and rice (LOC_Os01g72090; score of 332, similarity of 74.1%) (Figure 4B).

The changes in 3D structures between the *Gmeny* and WT Glyma.02G304700 protein can be clearly visualized when using the SWISS-MODEL online modeling server and PyMOL (Figure 4C,D).

Based on over 5000 RNA-seq data sets downloaded from http://soyatlas.venanciogroup.uenf.br (accessed on 26 December 2023), the three homologous genes *Glyma.02G304700*, *Glyma.14G009100*, and *Glyma.14G136300* displayed constitutively high expression in all tissues, especially in the shoot, seedling, leaf, and reproductive tissues, such as the flower and seed pod (Figure 4E). The expression patterns of *Glyma.02G304700* and *Glyma.14G009100* among the different tissues were particularly similar.

### 2.5. Haplotypes of Glyma.02G204700

A total of 1308 genomic sequences from 1308 cultivars were subjected to a haplotype analysis [26,27]. The genomic variants corresponding to the putative coding region of *Glyma.02G204800* were precluded. The resultant 10 single-nucleotide polymorphism (SNP) loci are listed in Appendix A. A haplotype possessed by two or more cultivars or accessions was retained for further analysis, resulting in 30 haplotypes (Appendix A).

According to the functional annotation, the C-to-T mutation at position 47,977,585 is the key allelic variation; the C-type haplotype was dominant in America (6/134), while the T-type haplotype is proportionally higher in Asia (319/982) and Europe (18/49), far higher than in America (6/131) (Appendix A). Also, the nonsynonymous C-to-G mutation at position 47,977,626 was detected in only one cultivar, ZYD3938 (PI549017), of Asian origin (Appendix A). The other eight SNPs are 3′UTR synonymous variants or 5′UTR variants. Also, there was no significant correlation between haplotypes with different maturity groups (MGs) (Appendix A).

### 2.6. Transcriptome Analysis

We next extracted RNA from leaf samples of *Gmeny* and the wild type, and the RNA quality was presented in Appendix A. RNA-seq analysis was performed to explore the diversified molecular functions of *Glyma.02G304700*, using the RNA-seq workflow pipeline (https://github.com/twbattaglia/RNAseq-workflow, accessed on 1 May 2020) (Figure 3). We manually assessed the authenticity of the genotype at the *Gmeny* locus for each sample on the IGV.

Of a total of 40,235 non-zero read counts, 7945 genes (20%) were significantly upregulated and 7393 genes (18%) were significantly downregulated in the mutant relative to the WT. A total of 777 genes (1.9%) (adjusted *p*-value < 0.05) were classified as low counts, while 25 genes (0.062%) were outliers. We identified the differentially expressed genes (DEGs) displaying a two-fold difference in expression between the WT and the *Gmeny* mutant in V4-stage leaves using ClueGO. A gene ontology (GO) enrichment analysis of these DEGs revealed the enrichment of 53 functions (Appendix A), including zeatin biosynthesis (eight DEGs); phenylpropanoid biosynthesis (30 DEGs); cutin, suberin, and wax biosynthesis (12 DEGs); the AGE-RAGE signaling pathway in diabetic complications (seven DEGs); DNA replication (29 genes); and mismatch repair (12 genes). The networks between these molecular functions are presented in Appendix A.

To test the reliability of the RNA-seq data, a real-time qPCR was performed. Eight DEGs (*Glyma.15G029500*, *Glyma.14G003200*, *Glyma.10G142600*, *Glyma.18G021500*, *Glyma.04G124300*, *Glyma.03G170300*, *Glyma.04G205600*, and *Glyma.11G003200*) with putative functions in the GA signaling pathway and in photosynthesis were randomly selected. *Glyma.15G029500*, *Glyma.03G170300*, *Glyma.04G205600*, and *Glyma.11G003200* were significantly upregulated in the *Gmeny* mutant, whereas the expression levels of *Glyma.14G003200*, *Glyma.10G142600*, *Glyma.18G021500*, and *Glyma.04G124300* were significantly downregulated in comparison with the WT. The RT-qPCR results were consistent with the trends revealed by the RNA-seq analysis, demonstrating the reliability of the RNA-seq analysis (Figure 5A–H).

Given that changes in *Glyma.02G304700* can lead to changes in leaf color and chlorophyll content, photosynthetic parameters, photomorphogenesis, and flowering time, we performed a detailed analysis of genes potentially involved in photosynthesis [28] and flowering time [29]. We used the 254 *Chlamydomonas reinhardtii* genes with potential functions in photosynthesis [28] as queries in a BLAST search of the soybean V275 genome, which led to the identification of 158 putative homologs. Of the 146 genes with a non-zero total read count, 39 were upregulated, including *Glyma.20G210400* (*RAA17*), *Glyma.17G052700* (*CPL12*), *Glyma.13G106600* (*CPL12*), *Glyma.07G027000* (*PSR17*), and *Glyma.09G187800* (*HCF101*) (Figure 5I, Appendix A). In addition, 41 genes were downregulated, including *Glyma.09G132300* (*HCF101*), *Glyma.16G179200* (*HCF101*), and *Glyma.17G205700* (*PIIR1*). *Glyma.12G030800* (*PIR2*), *Glyma.03G117500* (*PIR3*), *Glyma.07G109600* (*PIR3*), *Glyma.07G102100* (*PSR15*), *Glyma.17G097200* (*PSR15*), *Glyma.12G212000* (*PSR17*), *Glyma.03G161300* (*RAA12*), and *Glyma.04G193700* (*RAA17*) (Figure 5I, Appendix A).

Of the 701 potential flowering-related genes, including those involved in the GA pathway [29,30], 153 genes (22%) were downregulated in the mutant, such as those encoding TCP (TEOSINTE BRANCHED1, CYCLOIDEA, PCF) transcription factors, e.g., *Glyma.10G285900* and *Glyma.09G284300*, and others encoding SPL (SQUAMOSA PROMOTER BINDING PROTEIN LIKE) transcription factors, e.g., *Glyma.03G117600* and *Glyma.19G146000* (Figure 5J, Appendix A). A further 141 genes (20%) were upregulated, e.g., *Glyma.16G044100* (*FT*), *Glyma.19G260900* (*LHY*), *Glyma.01G023500* (*SVP*), *Glyma.02G121600* (*AP1*), *Glyma.13G052100* (*AGL*), and *Glyma.09G149000* (*AGL*) (Figure 5J, Appendix A). These changes in flowering time-related genes might be the cause of the early flowering and maturation phenotype of *Gmeny*.

In addition, the upregulation of the *PIF* gene *Glyma.18G115700* and GA-related genes *Glyma.17G178300* (*GA2ox8*) and *Glyma.04G211100* (*GA20ox2*) (Figure 5J, Appendix A) might be directly responsible for the elongated nodes during photomorphogenesis at the early vegetative growth stages.

## 3. Discussion

### 3.1. Mutation of a GmHY2a Gene Affects PΦB Biosynthesis, Inactivating the Photopigment System and Causing Diverse Changes in Photosynthesis, Photomorphogenesis, and Flowering Time

As shown in the transcriptome analysis, the mutation of the *GmHY2a* gene in *Gmeny* affects PΦB biosynthesis and consequently inactivates the entire photosensitive pigment system. The inactivation of phyB in particular stabilizes the PIFs, allowing them to bind to downstream target sites and exert their function. PIF1 regulates the expression of *PROTOCHLOROPHYLLIDE OXIDOREDUCTASE* (*POR*), *FERROCHELATASE* (*FeChII*), and *HEME OXYGENASE* (*HO3*) in the dark [31], while POR catalyzes the reduction of protochlorophyllide to chlorophyllide in magnesium branches under light. FeChII and HO3 catalyze the biosynthesis of heme from protoporphyrin IX and the conversion of heme to biliverdin IX-α in the iron branch, respectively (Figure 6). PIF3 inhibits the expression of *Glu tRNA REDUCTASE* (*HEMA1*), *GENOMES UNCOUPLED 4* (*GUN4*), and *Mg-CHELATASE SUBUNIT H* (*CHLH*) [25,32]. HEMA1 is the rate-limiting enzyme for the catalysis of ALA biosynthesis, while GUN4 and CHLH promote the shunting of ALA into the chlorophyll biosynthesis pathway [33].

In this study, we identified putative photosynthesis-related genes that were downregulated in the *Gmeny* mutant, including *Glyma.09G132300* (*HCF101*), *Glyma.16G179200* (*HCF101*), *Glyma.17G205700* (*PIIR1*), *Glyma.12G030800* (*PIR2*), *Glyma.03G117500* (*PIR3*), *Glyma.07G109600* (*PIR3*), *Glyma.07G102100*, *Glyma.17G097200* (*PSR15*), *Glyma.12G212000* (*PSR17*), *Glyma.03G161300* (*RAA12*), and *Glyma.04G193700* (*RAA17*) (Figure 5I, Appendix A). These DEGs might be associated with the decrease in chlorophyll contents and changes in photosynthetic performance in the *Gmeny* mutant (Figure 6).

In addition, the inactivation of phyB stabilizes PIF3, PIF4, and PIF5, enabling them to bind and activate downstream genes, including the genes involved in promoting stem elongation [34]. PIF activity is also influenced by the negative regulators of the shade-avoidance response, such as LONG HYPOCOTYL IN FAR RED 1 (HFR1), PHYTOCHROME RAPIDLY REGULATED 1 (PAR1), and PAR2. These proteins inhibit the transcriptional activity of the PIFs by interacting with their DNA-binding domain [35,36]. Similarly, FAR-RED ELONGATED HYPOCOTYL 3 (FHY3) and FAR-RED IMPAIRED RESPONSE 1 (FAR1) have also been shown to interact with PIF5 and PIF3 as negative regulators of the shade-avoidance responses. They can also directly activate the expression of *PAR1* and *PAR2* to downregulate the shade-avoidance response [37].

DELLA proteins, key components in the GA signaling pathway, can interact with PIFs and affect their activity. The transcription factors B-BOX DOMAIN PROTEIN 24 (BBX24) and BBX25 can interact with DELLAs, preventing their interaction with PIF4 and thereby lifting the inhibition of PIF4 activity, which promotes GA-induced cell elongation [18,38]. CONSTITUTIVE PHOTOMORPHOGENIC 1 (COP1), a central suppressor of photomorphogenesis, can target and degrade the transcription activator HFR1 to maintain PIF stability [34]. BIN2 can mediate the phosphorylation and degradation of PIF3 and PIF5, while COP1 can maintain the stability of PIN3 and PIN5 by inhibiting BIN2 [34,39]. In addition, COP1 can directly regulate the stability of DELLA proteins in response to shade [40].

In this study, the *PIF* homolog *Glyma.18G115700* and the GA-related genes *Glyma.17G178300* (*GA2ox8*) and *Glyma.04G211100* (*GA20ox2*) were upregulated in the *Gmeny* mutant (Figure 5J, Appendix A). This demonstrates that the mutation of the *GmHY2a* gene directly or indirectly affects the expression of key genes in the GA biosynthesis pathway, ultimately affecting the content of bioactive GA and possibly explaining the observed elongation of the hypocotyl and epicotyl in the *Gmeny* mutant.

The flowering time genes have been extensively studied since the successful cloning of the *E1* gene [41]. The model E1–FT2a/5a–MDEs stands as the central model for the flowering time gene network [42]. In this study, the putative homolog of the key gene, *FT5a* (*Glyma.16G044100*), was upregulated in the leaves as early as the V4 stage. The *LHY (Glyma.19G260900)*, *SVP* (*Glyma.01G023500)*, *AP1* (*Glyma.02G121600*), *AGL* homologs (*Glyma.13G052100* and *Glyma.09G149000*) were also upregulated (Figure 5J, Appendix A). These genes have been demonstrated to be the florigen signal, which is produced in leaves and transmitted to the shoot apical meristem to initiate flowering [43]. Some of the downregulated genes, such as the *TCP* homologs (*Glyma.10G285900* and *Glyma.09G284300*) and the *SPL* homologs (*Glyma.03G117600* and *Glyma.19G146000*) (Figure 5J, Appendix A), might function as flowering repressors [29]. There were significant changes in the expression of *E1b* (*Glyma.04G156400*; log2FoldChange = −4.2075 at *p* = 0.006538). However, no significant changes were observed in the expression of the *E1* and *E1La* putative homologs, which is slightly different from the previous report on the mutant of *GmHY2a* [14].

Notably, *Glyma.02G304800* was identified on the opposite strand to *Glyma.02G304700 within its fourth intron*. We excluded the corresponding variants for the haplotype analysis as there was no polymorphism correlation between *Gmeny*, the WT, and Hefeng 55 at this region.

Based on the C-to-G nonsynonymous mutation at position 47,977,626 on chromosome 2, in the first exon of *Glyma.02G304700*, the cultivars from America were mainly Hap_1 (C type), while the proportion of T type was higher in Asia (319/982) and in Europe (18/49) (Appendix A). This phenomenon might be attributed to the fact that the American cultivars may have originated from C-type Chinese cultivars or landraces, while the modern cultivars in Europe may have originated from a T-type mutation in Asian cultivars. A rare variant, the C-to-T mutation at position 47,977,585, was present only in the cultivar ZYD3938 (PI549017) of Asian origin (Appendix A).

Other than a few cultivars or accessions from Africa and Australia (Appendix A), a total of 999 cultivars from Asia were analyzed, and these displayed a diverse distribution of almost all haplotypes. A total of 49 cultivars were from Europe: 31 were Hap_1, 16 were Hap_2, 1 was Hap_3, and 1 was Hap_13. Of the 137 cultivars from America, 123 were Hap_1, 5 were Hap_2, and 9 were distributed among the other seven haplotypes (Appendix A).

Because we do not have data relating to chlorophyll content or photosynthetic parameters for all cultivars, we cannot assess the effects of the different haplotypes or specific allelic variations on these phenotypes; however, the functional effects of these variations on photosynthesis, photomorphogenesis, and flowering time would certainly warrant further investigation.

### 3.2. The BVF-IGV Pipeline Is Capable of Identifying the Causal Mutation for the Gmeny Mutant in One Step

The BVF-IGV pipeline was established in our lab and was used to successfully identify the causal mutations [24]. Initially, we also used MutMap to analyze our bulk resequencing data, but it yielded too many peaks spanning a relatively large genomic region [23]. One such peak was detected at the end of chromosome 2, where *Glyma.02G304700* is located. It would have been difficult to identify the candidate gene solely using MutMap as assessing each individual peak is time-consuming. With the BVF-IGV pipeline, we successfully identified *Glyma.02G304700* as the candidate gene for *Gmeny*. The authenticity of the candidate gene was verified by an examination of the variants between WT and *Gmeny* and an association study using the heterozygous *Gmeny*-derived segregating populations of the M_3_ and M_4_ generations, as well as using F_2_ populations derived from a cross of *Gmeny* × Hefeng55. In this study, we used bulk sequence data at both the DNA and RNA levels in the BVF-IGV pipeline, the results of which could be verified. The BVF-IGV pipeline may therefore facilitate gene cloning in mutant libraries, recombinant inbred line populations, or near-isogenic isoclines, especially in segregating populations.

Taken together, we confirmed that a 10 bp deletion in the first exon of *GmHY2a* is the leading causal factor for the hypocotyl elongation, early maturation, and changes in photosynthetic performance observed in the *Gmeny* mutant, due to its altered regulation of the expression of genes involved in photosynthesis, GA signaling, and flowering time. This study extends our knowledge of how FDBRs affect the biosynthesis and function of the phytochrome chromophore.

## 4. Materials and Methods

### 4.1. Mutant Library Generation and Phenotyping

Two mutant libraries were constructed by the 60Co-γ-ray-induced mutagenesis of a modern soybean cultivar, “Dongsheng 7”, from Northern China. Each line of the third generation of the mutant population (M_3_) derived from a single self-pollinated M_2_ plant was grown in the field for phenotypic observation. The mutant with elongated nodes and yellowing leaves was defined as *Gmeny*. The line T372 exhibited a distinct segregation between *Gmeny* and the wild type (WT).

### 4.2. Measurement of Photosynthetic Pigments and Parameters

New fully expanded leaves of the *Gmeny* mutant and “Dongsheng 7” were collected at three growth stages: the cotyledon stage (VC), the first trifoliolate stage (V1), and the third trifoliolate stage (V3). Three replicates of approximately 0.2 g of leaf tissue were steeped in 80% acetone for two days to extract the photosynthetic pigments. The total photosynthetic pigment content was determined using a UV/VIS spectrophotometer [44,45]. The contents of chlorophyll a, chlorophyll b, and carotenoid were also measured using a spectrophotometer, according to the previously described protocol [46], with slight modifications. The equations for the calculation of photosynthetic pigments were chlorophyll a (mg/L) = 9.784 OD_663_ − 0.990 OD_645_; chlorophyll b (mg/L) = 21.426 OD_645_ − 4.650 OD_663_; and carotenoids (mg/L) = 4.695 OD_440_ − 0.268 (chlorophyll a + chlorophyll b). The final contents were expressed as unit of mg/g of fresh weight.

The photosynthetic parameters (e.g., net photosynthetic rate, stomatal conductance, transpiration rate, and internal CO_2_ concentration) were measured from the third leaf down from the top of the *Gmeny* and “Dongsheng 7” plants at the V7 stage using the LI-COR 6800 Portable Photosynthesis System (LI-COR Biosciences, Lincoln, NE, USA), as described by Zhang et al. [47].

### 4.3. Transmission Electron Microscopy

An H-7650 transmission electron microscope (Hitachi, Tokyo, Japan) was used to investigate the ultrastructure of the chloroplast [48,49]. The leaf samples were collected from the same node of the *Gmeny* and “Dongsheng 7” plants at 20 days after germination in an experimental field in Harbin, Heilongjiang Province, China (45°41′ N, 126°38′ E).

### 4.4. Identification of the Causal Gene Using MutMap and BVF-IGV

The M_3_ generation of the T372 line was separated into two groups (bulks) based on phenotype. The mutant bulk comprised 12 *Gmeny* plants, while the WT bulk contained 28 WT plants. The genomic DNA and RNA of each bulk were extracted from leaf tissues of plants of the same size, and these samples were resequenced on an Illumina (San Diego, CA, USA) platform at Annoroad Gene Technology (Beijing, China). The reads were quality-trimmed using the NGS QC Toolkit (version 2.3.3) with default parameters [50]. The sequences of the two bulks were used to identify a causal region or peak using MutMap software with the default parameters [23]. Due to the limitations of MutMap revealed in this study, a manual BVF-IGV pipeline was subsequently developed, consisting of bulk sequencing, variant calling, functional annotation by SnpEFF, and batch integrative genomics viewer (IGV) observation for the identification of candidate genes [24] (Figure 3).

After the reads were quality-trimmed using the NGS QC Toolkit, the bulk sequences of the *Gmeny* mutants and WT plants were aligned to the reference genome Gmax_275_Wm82.a2.v1 (V275) using SpeedSeq [51]. Variants were obtained from BAM files using GATK version 2.3-3 with the parameters -stand_emit_conf 10 and -stand_call_conf 30. The VCF file was filtered using the FilterVcf function with parameters MIN_AB = 0.8, MIN_DP = 6, MIN_GQ = 0, and MIN_QD = 2. The resulting VCF file was functionally annotated using SnpEFF with default parameters [52] (Figure 3).

### 4.5. Generation of the F_2_ Population and Association Analysis

To verify the identification of the causal gene underlying the *Gmeny* mutation performed using BVF-IGV, *Gmeny* was crossed with Hefeng 55 to generate a segregating population. The segregation ratio of the *Gmeny* and WT phenotypes was analyzed, and association studies between the phenotype and genotype were conducted to confirm the causal mutation.

### 4.6. Multiple-Sequence Alignment and Phylogenetic Analysis

Proteins encoded by homologs of *Glyma.02G304700* (used as a query) were identified in the Phytozome database (http://www.phytozome.net, accessed on 12 December 2023) for various plant species, including *Arabidopsis thaliana*, rice, common bean (*Phaseolus vulgaris* L.), *Medicago truncatula*, mung bean (*Vigna radiata* (L.) R. Wilczek), bitter blue lupin (*Lupinus micranthus* Guss.), and soybean. Multiple-sequence alignments of the amino acid sequences for the PΦB synthase proteins were performed using ClustalX (version 2.0.9) [53], and the phylogenetic tree was constructed in MEGA 7.0 using the neighbor-joining method [54].

### 4.7. Haplotype Analysis

A total of 1308 genomic sequences from 1308 cultivars were included in this analysis [26,27]. All variants within the *Glyma.02G304700* gene region were retrieved first. A relatively large proportion of SNPs corresponding to *Glyma.02G304800* were located on the plus strand of the fourth intron of *Glyma.02G304700* were excluded before further analysis. Haplotypes were classified based on the variation profiles of 10 SNP loci. The geographic origin and proportion of the haplotypes were visualized using Popart (version 1.7) [55].

### 4.8. RNA Sequencing and Transcriptome Analysis

Leaf samples were collected from the same node of “Dongsheng 7” and *Gmeny* for transcriptome sequencing. Three replicates of each genotype were sent to Annoroad Gene Technology Corporation (Beijing, China) for RNA sequencing (RNA-seq). The RNA-seq data were analyzed through the RNA-seq workflow pipeline (https://github.com/twbattaglia/RNAseq-workflow, accessed on 1 May 2020) against the soybean reference genome (V275 of Wm82.a2.v1; https://phytozome-next.jgi.doe.gov, accessed on 23 April 2016). The differential gene expressions (DGEs) were further analyzed using ClueGo and cytoHubba to annotate their molecular functions [56,57], and the enriched functional pathways were visualized using Cytoscape version 3.9.1 [58].

### 4.9. Quantitative RT-PCR

A quantitative RT-PCR (qRT-PCR) analysis was performed to validate the RNA-seq results. The total RNA was extracted from the different tissues using an OmniPlant RNA Kit (DNase I) (CW25985; CWBIO, Taizhou, Jiangsu, China). A total of 500 ng of RNA was subjected to reverse transcription using TransScript One-Step gDNA Removal and cDNA Synthesis SuperMix (AT311-03; TransGen Biotech, Beijing, China). TransStart Top Green qPCR SuperMix (AQ131-04; TransGen Biotech) was used for the qRT-PCR analysis. The qRT-PCR reaction was performed on a LightCycler 96 (Roche, Basel, Switzerland). The measured Ct values were converted to relative copy numbers using the −∆∆Ct method [59].

### 4.10. Statistical Analysis

The Student *t*-test, analysis of variance (ANOVA), and chi-square test were performed using GraphPad Prism version 9.5.1 (GraphPad, San Diego, CA, USA).

## Figures and Tables

**Figure 1 ijms-25-06483-f001:**
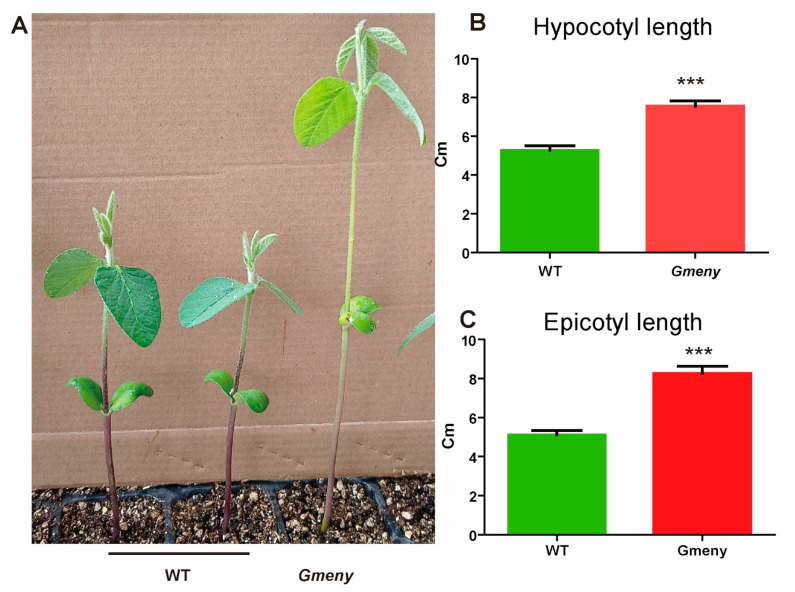
The *Gmeny* mutant displays yellowing leaves and elongated nodes at the vegetative stage. (**A**) Phenotypes. (**B**) Hypocotyl lengths. (**C**) Epicotyl lengths. *** represents Student *t*-test at *p* < 0.001. Error bar represents standard deviation.

**Figure 2 ijms-25-06483-f002:**
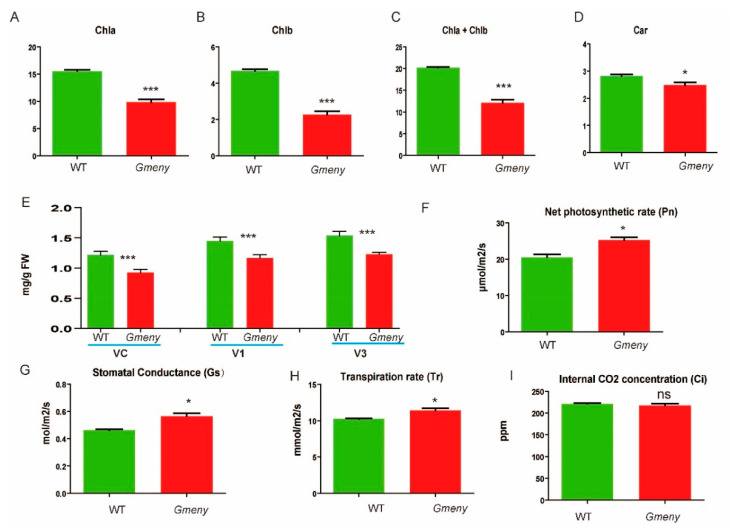
Chlorophyll contents and physiological and anatomical differences in photosynthesis between the *Gmeny* mutant and the wild type (WT). (**A**) Chlorophyll a contents. (**B**) Chlorophyll b contents. (**C**) Total chlorophyll a + chlorophyll b contents. (**D**) Carotenoid contents. (**E**) Total pigment contents at three growth stages. (**F**–**I**) Photosynthetic parameters of plants grown in the field. (**F**) Net photosynthetic rate. (**G**) Stomatal conductance. (**H**) Transpiration rate. (**I**) Internal CO_2_ concentration. (**J**–**O**) Transmission electron microscopy images showing the differences in chloroplast structure between the WT (**J**–**L**) and *Gmeny* mutant (**M**–**O**). OB: osmiophilic body; SG: starch granules; and Thy: thylakoid membrane. Bars = 2 μm; *, *** represent Student *t*-test at *p* < 0.05, and 0.001, respectively; ns, no significant difference. Error bar represents standard deviation.

**Figure 3 ijms-25-06483-f003:**
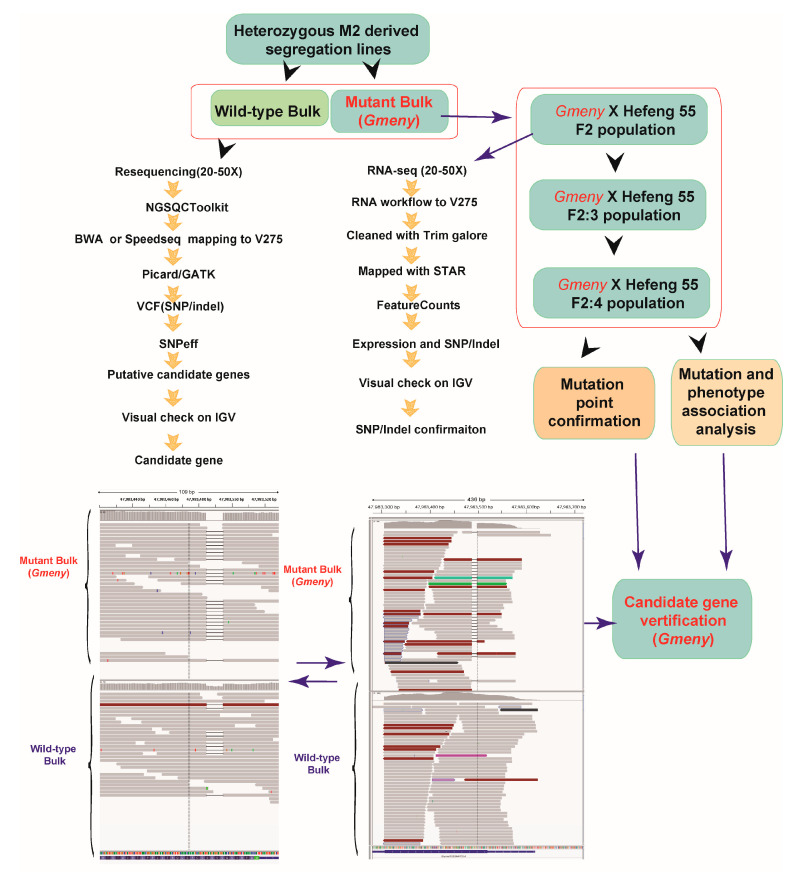
Strategic approach for the identification and confirmation of the candidate gene mutated in the *Gmeny* mutant using the BVF-IGV pipeline. First, the bulk resequencing data were processed through the BVF-IGV pipeline, which includes bulk sequencing, variant calling, functional annotation using SnpEFF software (v4.0e), and the Integrative Genomics View [24,25]. The bulk transcriptome data were also analyzed using the BVF-IGV pipeline. The results of the bulk resequencing data and the transcriptome data were also used to verify each other. An F_2_ population and their derived sub-populations were made from a cross between *Gmeny* × Hefeng 55 to verify the authenticity of *Glyma.02G304700* as the candidate predicted by BVF-IGV.

**Figure 4 ijms-25-06483-f004:**
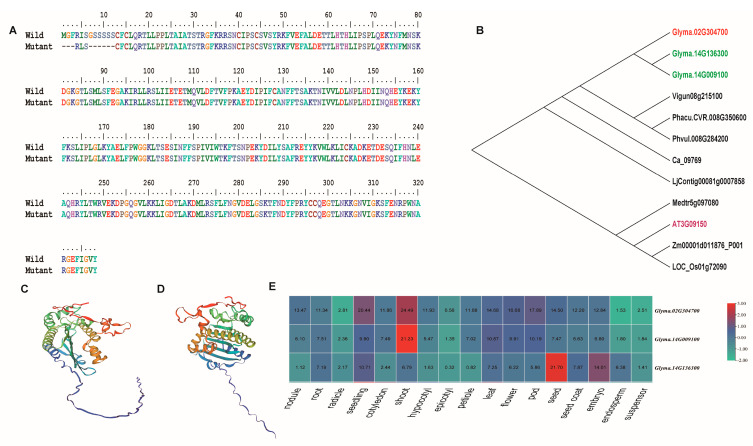
Characterization of the candidate gene, *Glyma.02G304700*. (**A**) Alignment of the wild-type (WT) and *Gmeny* mutant Glyma.02G304700 proteins. The 10 bp deletion in the first exon of *Glyma.02G304700* leads to a 12-amino acid change (9 amino acids shorter in the *Gmeny* mutant). (**B**) Phylogenetic tree of Glyma.02G304700 protein sequences from leguminous species and two model plants, *Arabidopsis thaliana* and rice (*Oryza sativa*). All sequences were retrieved from Phytozome (v13) and were aligned using the ClustalW program in the Bioedit software. (**C**,**D**) Predicted 3D protein structures for Glyma.02G304700 in the WT (**C**) and *Gmeny* mutant (**D**) using a SWISS-MODEL online modeling server and visualized with PyMOL. (**E**) The expression of *Glyma.02G304700* and its homologous genes in different soybean tissues. The expression data were downloaded from http://soyatlas.venanciogroup.uenf.br (accessed on 26 December 2023).

**Figure 5 ijms-25-06483-f005:**
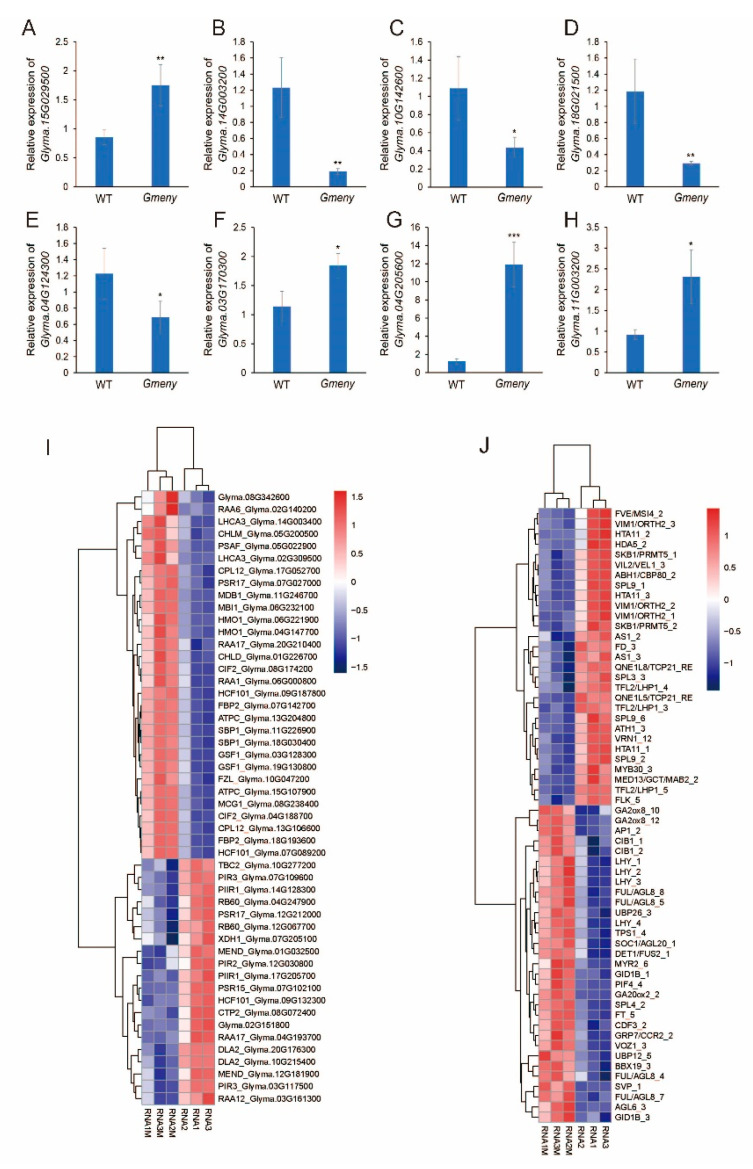
Gene expression profiles and heat maps of the genes involved in photosynthesis and flowering time regulation. (**A**–**H**) Expression of selected genes determined using RT-qPCR to validate the RNA sequencing results. WT: wild type. (**A**) *Glyma.15G029500*. (**B**) *Glyma.14G003200*. (**C**) *Glyma.10G142600*. (**D**) *Glyma.18G021500*. (**E**) *Glyma.04G124300*. (**F**) *Glyma.03G170300*. (**G**) *Glyma.04G205600*. (**H**) *Glyma.11G003200*. (**I**,**J**) Heat maps of the expression levels of genes involved in photosynthesis (**I**) and flowering time (**J**). *, **, and *** represent Student *t*-test at *p* < 0.05, 0.01, and 0.001, respectively. Error bar represents standard deviation.

**Figure 6 ijms-25-06483-f006:**
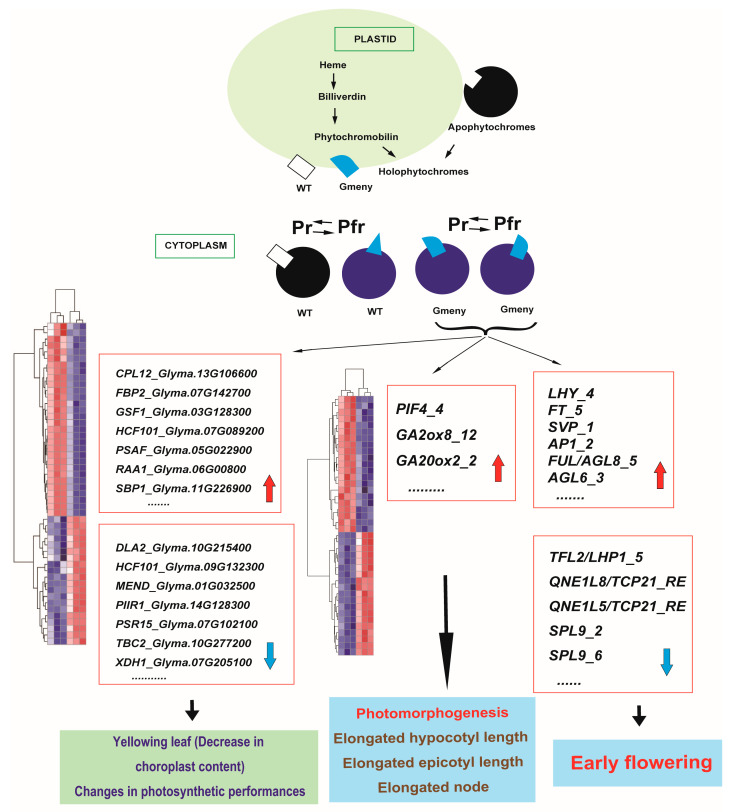
Diagram of the possible functional mechanism of GmHY2a. In plastids, the *Gmeny* mutant and wild-type (WT) GmHY2a proteins are structurally different. When transferred into the cytoplasm, the WT and *Gmeny* proteins differed in their shifts between the Pr and Pfr forms, which might result in the dynamic differences in photosynthesis, photomorphogenesis, and flowering time observed in these genotypes due to differences in the regulation of downstream gene expression levels. Red arrows represent the up-regulation, while blue arrows stand the down-regulation.

## Data Availability

The original contributions presented in this study are publicly available. The raw sequence data reported in this article have been deposited in the Genome Sequence Archive (Genomics, Proteomics & Bioinformatics 2021) at the National Genomics Data Center (Nucleic Acids Res 2021), China National Center for Bioinformation/Beijing Institute of Genomics Chinese Academy of Sciences (subPRO037650), and are publicly accessible at https://ngdc.cncb.ac.cn/gsa (accessed on 26 December 2023).

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
