# Peer review of "A Novel 10-Base Pair Deletion in the First Exon of GmHY2a Promotes Hypocotyl Elongation, Induces Early Maturation, and Impairs Photosynthetic Performance in Soybean"

_ijms, 2024, doi:10.3390/ijms25126483_

Round 1

Reviewer 1 Report

Comments and Suggestions for Authors

The work “A novel 10-base pair deletion at the first exon in GmHY2a promotes hypocotyl elongation, early-maturing, and impairs photosynthetic performance” describes the mutant characterization of a soybean Gmeny mutant (i.e., gmhy2a gene mutant; gamma-induced on Dongsheng, a gene that encodes PΦB synthase, which in turn is a chromophore for plant phytochromes). These mutant materials are characterized by long internodes, affected chlorophyll synthesis and early flowering phenotype. Understanding mutations on this synthase contribute to the knowledge of phytochrome function, providing insights into plant growth, development, and adaptation. In a long-term meaning, these studies could contribute to understand and enhance crop yield and performance, by increasing plant efficiency under designed environments. In that line, the work covers eventually relevant areas in plant improvement, from molecular to practical approaches.

Introduction section. Whereas the section incorporates a good amount of relevant information and conducts us into the relevance of these gmhy2a (phytochromobilin synthase gene) mutations, it draws attention that a previous description by Zhang et al. (2022; current ref #5) is not further considered in these terms but for other purposes, as seen in line 152. In addition, in the section ref #5 is mentioned for an Arabidopsis context (line 60); however, it is a quite related article (GMlin1 mutant; EMS-induced in Williams 82) to the current one, considering both works are describing mutation in Glyma.02G30470. Please, clarify.

I´m going to Results because comments for this section will mean on Methods.

Results section. The section presents good starting sub-sections, showing an organization that allows to the reader to follow and understand relevant phenotype findings/conditions in the mutant. However, after these, the work felt into a more tangled up delivery of the obtained data. For instance, at “The BVF-IGV pipeline identifying Glyma.02G304700…”, methods are mixed at the time that NGS results are given (lines 277-282). A good summary Table for NGS results should be OK instead of the wordy paragraph. The same for the next paragraph (lines 284-295), in which several packages and setting are indicated or compared, instead of a direct delivery of the best obtained/deduced results.  In addition, is Figure 3 (strategic approach for identification…) falling into Results or useful in Methods?

I strongly suggest that generating a solid description of the lab-made pipeline (BVF-IGV) to be considered in Materials-Methods, and later just investing efforts in giving Results for what was generated by that one, it will improve the work presentation, fluidity and understanding. Missing some sections at this rate will not mean losing hierarchy. This is valid for sections in lines 322-347).

Considering the purpose of the work, what is the relevance of incorporating haplotype analyses for this gene and mutant? (lines 386-406). May be this paragraph (sub-section, and even Figure 5) can be useful in Discussion and these results under Supplementary data format). Which are the criteria to go into haplotypes when an induced mutant is generated? Is this relevant to the main context of the study?

Transcriptome analysis. Whereas Tables are not a preferred way to deliver results sometimes, however, that format will allow to give us a deeper and (visually) concentrated understanding of the meaning of Gmeny mutant.

Figure 6 is not helping to any reader, please, select main or relevant pathways. Also, creating Supplementary material at this rate will help you keeping al of the information you want to deliver us. In the same line, for instance, considering the qRNA seq selection, an illustration about networks for those included genes should result much more of interest. And may be just focusing this area of the work just in photosynthesis and flowering networks, considering the next paragraphs in the manuscript. I think this is quite relevant considering other mutants described for the same locus (GMlin1 by Zhang et al., 2022 ref #5).

Others in the section:

Line 224: AI editor / checker?

Line 276: Dnsheng

Line 296: “= 2 μm).” (???)

Line 410: “TableS?”

Methods.

In general, methods should be conveniently organized according to results, this is because may be some sections in results will be reformulated according to suggestions.

-Reseq and identification (lines 158 to 170). Multiple paragraphs in Results sections should be included in this section to improve organization in that section. Se comments below.

 Discussion. The very first question to be considered in the discussion is which is the relevance and novelty of the Gmeny? This is (quite) relevant considering other mutants with the same phenotype. In that line, why authors decided pursuing this mutant if other similar materials have been generated alreadyAre lines 510-512 news for the community? (these are questions that (may be) the reader wants to get at this rate of the work).  

Figure 8 (and texts about it) are attractive; and for instance, were these genes included in qPCR analyses? I think this should be the focus conducting (building) Results descriptions.

As indicated previously, which is the relevance of considering population analyses on the basis of a mutant generated experimentally? Authors need to consider that keeping focus on main issues in the work could have a much more valuable message. I accept that haplotypes versus flowering time versus mutant and locations are quite interesting; however I don´t know if this is the conducting idea in the current presentation of the work (I see it some confusing yet).

Comments on the Quality of English Language

After reviewing this current version, authors should carefully check language issues. 

Author Response

The work “A novel 10-base pair deletion at the first exon in GmHY2a promotes hypocotyl elongation, early-maturing, and impairs photosynthetic performance” describes the mutant characterization of a soybean Gmeny mutant (i.e., gmhy2a gene mutant; gamma-induced on Dongsheng, a gene that encodes PΦB synthase, which in turn is a chromophore for plant phytochromes). These mutant materials are characterized by long internodes, affected chlorophyll synthesis and early flowering phenotype. Understanding mutations on this synthase contribute to the knowledge of phytochrome function, providing insights into plant growth, development, and adaptation. In a long-term meaning, these studies could contribute to understand and enhance crop yield and performance, by increasing plant efficiency under designed environments. In that line, the work covers eventually relevant areas in plant improvement, from molecular to practical approaches.

Author’s response: Thank you for your comments.

Introduction section. Whereas the section incorporates a good amount of relevant information and conducts us into the relevance of these gmhy2a (phytochromobilin synthase gene) mutations, it draws attention that a previous description by Zhang et al. (2022; current ref #5) is not further considered in these terms but for other purposes, as seen in line 152. In addition, in the section ref #5 is mentioned for an Arabidopsis context (line 60); however, it is a quite related article (GMlin1 mutant; EMS-induced in Williams 82) to the current one, considering both works are describing mutation in Glyma.02G30470. Please, clarify.

Author’s response: Thank you for careful check. Indeed, the ref#5 is very relevant to the current study, although focus of ref#5 is different from the current study. Zhang et al., (2022) (ref. 5) focused on the influence of light quality, e.g. both red and far-red light on de-etiolation response and shade avoidance. Although Zhang et al., (2022) revealed E1 might be the downstream genes for GmHY2a, which is different from what we found in this study. Therefore, the results and conclusion of this study will further strength the function of this gene in several new aspects. Mutation of a GmHY2a gene affects PΦB biosynthesis, inactivating the photopigment system and causing diverse changes in photosynthesis, photomorphogenesis, and flowering time

As to “In addition, in the section ref #5 is mentioned for an Arabidopsis context (line 60)” , we applogy for the wrong citation, and we corrected it in the new version.

As to is not further considered in these terms but for other purposes, as seen in line 152.

Since we defined this mutant earlier than Zhang’s publication, and we think this term is quite correctly reflect the characterization of the mutant, therefore, we did not follow the terms that was proposed in Zhang et al., (2022). However, IN the line 152, we described the metholodgy for measuring photosynthetic parameters using the protocol of Zhang et al., (2022) (ref 28), which is different from Zhang et al., (2022)(ref 5).  

I´m going to Results because comments for this section will mean on Methods.

Results section. The section presents good starting sub-sections, showing an organization that allows to the reader to follow and understand relevant phenotype findings/conditions in the mutant. However, after these, the work felt into a more tangled up delivery of the obtained data. For instance, at “The BVF-IGV pipeline identifying Glyma.02G304700…”, methods are mixed at the time that NGS results are given (lines 277-282). A good summary Table for NGS results should be OK instead of the wordy paragraph. The same for the next paragraph (lines 284-295), in which several packages and setting are indicated or compared, instead of a direct delivery of the best obtained/deduced results.  In addition, is Figure 3 (strategic approach for identification…) falling into Results or useful in Methods?

I strongly suggest that generating a solid description of the lab-made pipeline (BVF-IGV) to be considered in Materials-Methods, and later just investing efforts in giving Results for what was generated by that one, it will improve the work presentation, fluidity and understanding. Missing some sections at this rate will not mean losing hierarchy. This is valid for sections in lines 322-347).

Author’s response: Thank you for your kind suggestion. We have reorganized the relevant parts by moving all the methodology into the section of “Materials and Methods”. Also we deleted some irrelevant description. We think all introduction, materials and methods, result and discussion look more logic in the revised version.

Considering the purpose of the work, what is the relevance of incorporating haplotype analyses for this gene and mutant? (lines 386-406). May be this paragraph (sub-section, and even Figure 5) can be useful in Discussion and these results under Supplementary data format). Which are the criteria to go into haplotypes when an induced mutant is generated? Is this relevant to the main context of the study?

Author’s response: Thank you for your kind suggestion. Haplotype analysis helps to understand the genetic structure and variation of a population, providing important insights into population genetics and evolution. By analysing haplotypes, it is possible to identify specific genetic variants associated with traits, which is beneficial for functional study. In order to apply the discovering of this study to soybean production or breeding, the excellent haplotype is good and easy to be used. Also the haplotype analysis can help us to track the domestication process. Indeed, due to there no phenotypic data relevant to the photosynthesis or other traits e.g. internode length, we might not be able to predict which haplotype is functional active or dysfunctional. Per your suggestion, we keep minimal part of haplotype in the result section and discussed a little more in the section of discussion part.

Transcriptome analysis. Whereas Tables are not a preferred way to deliver results sometimes, however, that format will allow to give us a deeper and (visually) concentrated understanding of the meaning of Gmeny mutant.

Figure 6 is not helping to any reader, please, select main or relevant pathways. Also, creating Supplementary material at this rate will help you keeping al of the information you want to deliver us. In the same line, for instance, considering the qRNA seq selection, an illustration about networks for those included genes should result much more of interest. And may be just focusing this area of the work just in photosynthesis and flowering networks, considering the next paragraphs in the manuscript. I think this is quite relevant considering other mutants described for the same locus (GMlin1 by Zhang et al., 2022 ref #5).

Author’s response: As you pointed out, some irrelevant materials might dilute the interesting point of this paper. Therefore, we have moved some relatively less important picture into the supplemental materials, and focused on presentation of photosynthesis and flowering networks. In this study, in order to check the result of transcriptome analyses, the genes were randomly selected. Since the current work might not enable us to build a gene network for the gene listed in the Figure 6, we will further consider your suggestion in the further research on this subject.

Others in the section:

Line 224: AI editor / checker?

Line 276: Dnsheng

Line 296: “= 2 μm).” (???)

Line 410: “TableS?”

Author’s response: Thank you for your careful editing, we have revised all above accordingly.

Methods.

In general, methods should be conveniently organized according to results, this is because may be some sections in results will be reformulated according to suggestions.

-Reseq and identification (lines 158 to 170). Multiple paragraphs in Results sections should be included in this section to improve organization in that section. Se comments below.

Author’s response: Thanks for your suggestion. As you pointed out, some methodology related descriptions are tangle with the result, therefore, we have reorganized by moving all relevant paragraphs or descriptions into the section of Materials and Methods. 

 Discussion. The very first question to be considered in the discussion is which is the relevance and novelty of the Gmeny? This is (quite) relevant considering other mutants with the same phenotype. In that line, why authors decided pursuing this mutant if other similar materials have been generated alreadyAre lines 510-512 news for the community? (these are questions that (may be) the reader wants to get at this rate of the work).  

Author’s response: Thank your advice. We have changed the order in the discussion section with focus on the main points of this paper. Although the mutation of this gene was reported, however, previous work focused on the influence of light quality, e.g. both red and far-red light on de-etiolation response and shade avoidance. In this study, we concluded that Mutation of the GmHY2a gene will affect PΦB synthesis, leading to the inactivation of the entire photosensitive pigment system, thus resulting in diversified changes in photosynthesis, photomorphogenesis, and flowering time. We think the current work will deepen our knowledge on the function of GmHY2a.

Figure 8 (and texts about it) are attractive; and for instance, were these genes included in qPCR analyses? I think this should be the focus conducting (building) Results descriptions.

Author’s response: Thank you for your good comments and suggestion. Since the genes were selected randomly to show the correlation between RNA-seq and qRT-PCR. The result showed that the RNA-seq data are consistent with qRT-PCR. And the result showed the consistence between result of RNA-seq and the qRT-PCR.

Also, several crucial genes were selected, e.g.

 Glyma.14G003200      K00228   GO:0004109,GO:0006779,GO:0055114     ATCPO-I,HEMF1,LIN2       Coproporphyrinogen III oxidase

Glyma.10G142600        GO:0046983   PAP3,PIF3,POC1  phytochrome interacting factor 3

Glyma.18G021500        GO:0004853,GO:0006779    HEME2   Uroporphyrinogen decarboxylase

Glyma.04G124300        GO:0003677,GO:0004803,GO:0006313,GO:0008270      FHY3      far-red elongated hypocotyls 3

Glyma.03G170300        GO:0046983   PIF1,PIL5      phytochrome interacting factor 3-like 5

Glyma.04G205600        GO:0004325,GO:0006783    ATFC-II,FC-II,FC2 ferrochelatase 2

We will further follow your suggestion when we do further research on this subject.

As indicated previously, which is the relevance of considering population analyses on the basis of a mutant generated experimentally? Authors need to consider that keeping focus on main issues in the work could have a much more valuable message. I accept that haplotypes versus flowering time versus mutant and locations are quite interesting; however I don´t know if this is the conducting idea in the current presentation of the work (I see it some confusing yet).

Author’s response: Since currently we do not have enough phenotypic data eg. Chl contents, and photosynthetic parameters, therefore, we minimized relevant haplotype part in the result sanction and discussed a little more in the discussion section.

After reviewing this current version, authors should carefully check language issues. 

Author’s response: The revised version hs been edited by a profession service (Plant Editors).

Reviewer 2 Report

Comments and Suggestions for Authors

The paper was focused on the impact of a novel 10-base pair deletion at the first exon in GmHY2a on the hypocotyl elongation, early-maturing, and impairs photosynthetic performance. Gmeny mutant displaying elongated nodes and yellowing leaf was identified from a γ ray mutant library of cultivar 'Dongsheng 7'. The Authors identified a 10 bp deletion in the first exon of Glyma.02G304700. Furthermore, the authenticity of this mutation for Gmeny was verified by variation analysis of over 500 genes in the candidate gene region and association analysis in two types of populations segregated for Gmeny. Transcriptome analysis revealed diversified changes in multiple KEGG pathways. Altered gene expressions related to photosynthesis, GA signaling, and flowering time may be associated with physiological and phenotypical changes. The novel 10-base pair deletion at the first exon in Glyma.02G304700promotes hypocotyl elongation, early-maturation, and impairs photosynthetic performance.

In  my opinion, the paper is well organized, planned, and presents the interesting scientific results. However, I recommend some improvements:

-            The Introduction is too long, and overloaded in detailed information, hence, it should be presented in a more concise form.

-            I recommend including the electropherograms presenting the RNA bands in agarose gels in the Supplementary file – it would provide information regarding quality of total RNA samples. In addition, RIN (RNA integrity number) should be measured and included in the manuscript or in the Supplementary file.

-            If Authors used SYBR Green fluorescent dye during RT-PCR gene expression studies, hence, the results of Melting Curve Analysis should be added in the manuscript or Supplementary file (e.g., JPG or TIFF file).

-            There is the lack of citation of −∆∆Ct method used during RT-PCR gene expression analyses.

-            A subsection of Statistical analyses should be included.

-            There is also missing the name of the statistical test under the figures.

-            Moderate editing of English language required.

Comments on the Quality of English Language

-    Moderate editing of English language required.

Author Response

The paper was focused on the impact of a novel 10-base pair deletion at the first exon in GmHY2a on the hypocotyl elongation, early-maturing, and impairs photosynthetic performance. Gmeny mutant displaying elongated nodes and yellowing leaf was identified from a γ ray mutant library of cultivar 'Dongsheng 7'. The Authors identified a 10 bp deletion in the first exon of Glyma.02G304700. Furthermore, the authenticity of this mutation for Gmeny was verified by variation analysis of over 500 genes in the candidate gene region and association analysis in two types of populations segregated for Gmeny. Transcriptome analysis revealed diversified changes in multiple KEGG pathways. Altered gene expressions related to photosynthesis, GA signaling, and flowering time may be associated with physiological and phenotypical changes. The novel 10-base pair deletion at the first exon in Glyma.02G304700promotes hypocotyl elongation, early-maturation, and impairs photosynthetic performance.

In  my opinion, the paper is well organized, planned, and presents the interesting scientific results. However, I recommend some improvements:

Author’s response: Thank you for your comments.

-            The Introduction is too long, and overloaded in detailed information, hence, it should be presented in a more concise form.

Author’s response: Thank you for your suggestion, we have shortened this part and make it concise, also revised version had been edited by a profession service (Plant Editors).

-        In addition, RIN (RNA integrity number) should be measured and included in the manuscript or in the Supplementary file.

The RIN (RNA integrity number) regarding the quality of RNA for RNA-seq is presented in supplemental Table ()

Author’s response: Thank you for your advice, we prepare the RIN data and added in the Figure S .

    I recommend including the electropherograms presenting the RNA bands in agarose gels in the Supplementary file – it would provide information regarding quality of total RNA samples.

-            If Authors used SYBR Green fluorescent dye during RT-PCR gene expression studies, hence, the results of Melting Curve Analysis should be added in the manuscript or Supplementary file (e.g., JPG or TIFF file).

Author’s response: Since the qRT-PCR is routinely used in the lab, here is the image of gel and melting curves for some genes. Since in this paper, we did not enclosed in the main text or supplemental data, we provide the gel image here showing the quality of RNA extracted. Also we provide several melting curves for several genes showing the original data for your reference.

rna gel

The gel image showing the quality of RNA extracted.

qPCR dissolution curve (The figure above is the qPCR obtained in mutant and wild-type samples using the TUA5 as reference gene; the figure below is the qPCR obtained in mutant and wild-type samples using quantitative primers of randomly selected genes with significant up/down in RNAseq)

Glyma.15G029500

Glyma.14G003200

Glyma.10G142600

Glyma.18G021500

Glyma.04G124300

Glyma.03G170300

Glyma.04G205600

Glyma.11G003200

-            There is the lack of citation of −∆∆Ct method used during RT-PCR gene expression analyses.

Also we cited the method”The measured Ct values were converted to relative copy numbers using the −∆∆Ct method” (Schmittgen and Livak, 2008).

-            A subsection of Statistical analyses should be included.

-            There is also missing the name of the statistical test under the figures.

(done) Author’s response: we have included a paragraph for Statistical analyses

Statistical analysis

The Student’s t-test, analysis of variance (ANOVA), and chi-square test were performed using GraphPad Prism version 9.5.1 (GraphPad, San Diego, California, USA).

Also we have included missing the name of the statistical test for each figures, e.g. . *, **, *** Student t test at p < 0.05 and 0.01, respectively.

-            Moderate editing of English language required.

Author’s response:The revised version had been edited by a profession service (Plant Editors).
